# A comprehensive dataset of microbial abundance, dissolved organic carbon, and nitrogen in Tibetan Plateau glaciers

Yongqin Liu[1,2], Pengcheng Fang[3], Bixi Guo[2], Mukan Ji[1], Pengfei Liu[1], Guannan Mao[2], Baiqing Xu[2], Shichang Kang[4], Junzhi Liu[1*]

[1]Center for the Pan-Third Pole Environment, Lanzhou University, Lanzhou, 730000, China
[2]State Key Laboratory of Tibetan Plateau Earth System, Resources and Environment, Institute of Tibetan Plateau Research, Chinese Academy of Sciences, Beijing, 100101, China
[3]Key Laboratory of Virtual Geographic Environment (Nanjing Normal University), Ministry of Education, Nanjing, 210023, China
[4]State Key Laboratory of Cryospheric Science, Northwest Institute of Eco-Environment and Resources, Chinese Academy of Sciences, Lanzhou 730000, China

*Correspondence to*: Junzhi Liu (liujunzhi@lzu.edu.cn)

**Abstract.** Glaciers are recognized as a biome dominated by microorganisms and a reservoir of organic carbon and nutrients. Global warming remarkably increases glacier melting rate and runoff volume, which have significant impacts on the carbon and nitrogen cycles in downstream ecosystems. The Tibetan Plateau (TP), dubbed "the water tower of Asia", owns the largest mountain glacial area at mid- and low-latitudes. However, limited data on the microbial abundance, organic carbon, and nitrogen in TP glaciers are available in the literature, which severely hinders our understanding of the regional carbon and nitrogen cycles. This work presents a new dataset on microbial abundance, dissolved organic carbon (DOC), and total nitrogen (TN) for TP glaciers. In this dataset, there are 5409 records from 12 glaciers for microbial abundance in ice cores and snow pits, and 2532 records from 38 glaciers for DOC and TN in the ice core, snow pit, surface ice, surface snow, and proglacial runoff. These glaciers are located across diverse geographic and climatic regions, where the multiyear average air temperature ranges from -13.4 ℃ to 2.9 ℃ and the multiyear average precipitation ranges from 76.9 mm to 927.8 mm. This makes the constructed dataset qualified for large-scale studies across the TP. To the best of our knowledge, this is the first dataset of microbial abundance and TN in TP glaciers and also the first dataset of DOC in ice cores of the TP. This new dataset provides important information for studies on carbon and nitrogen cycles in glacial ecosystems, and is especially valuable for the assessment of potential impacts of glacier retreat on downstream ecosystems under global warming. The dataset is available from the National Tibetan Plateau/Third Pole Environment Data Center (https://doi.org/10.11888/Cryos.tpdc.271841, Liu, 2021).

## 1 Introduction

The glacier is regarded as an extreme environment, featured by the sustained low temperature, lack of nutrients, and strong radiation. Abundant and active microorganisms play important roles in the biogeochemical cycling in glacier ecosystems (Liu et al., 2016a; Smith et al., 2017; Irvine-Fynn et al., 2021), and the glacier is a reservoir of microorganisms, organic carbon, and nitrogen (Anesio and Laybourn-Parry, 2012; Hood et al., 2015). Carbon and nitrogen in glaciers could be originated from atmospheric deposition or microbial activities (Anesio et al., 2017; Hodson et al., 2008). These nutrients are vital for downstream aquatic ecosystems (Hodson et al., 2008; Dubnick et al., 2017), especially for the oligotrophic proglacial streams and lakes (Hood et al., 2015). Glaciers are also the species pools for downstream aquatic ecosystems and substantially influence the microbial functions in downstream ecosystems. It has been reported glacier originated bacteria can contribute up to 20% of the microbial community in the proglacial stream 20 km downstream (Liu et al., 2021), and glacier retreat can change the fungal abundance and cellulose decomposition rate in mountain rivers (Fell et al., 2021). Under global climate warming, glaciers are melting at an accelerating rate, the carbon, nitrogen, and microorganisms stored in glaciers will be released in glacier runoff, and may alter the carbon and nitrogen cycling in downstream ecosystems (Hood et al., 2009; Singer et al., 2012; Fellman et al., 2015; Wadham et al., 2016).

The Tibetan Plateau (TP), dubbed "the water tower of Asia", owns the largest mountain glacial area at mid- and low-latitudes (Yao et al., 2012). Glaciers on the TP are the sources of several large rivers in Asia and are thus of great importance for downstream regions (Yao et al., 2019). However, due to harsh climate conditions and logistic difficulties, there have been limited reports of microorganisms, organic carbon, and nitrogen in TP glaciers, especially in ice cores. Microbial abundance in ice cores is only available in three TP glaciers (i.e. Puruoganri, East Rongbuk, Malan; Yao et al., 2006; Zhang et al., 2008, 2010) using the epifluorescence microscopy method, and there is no publicly-available dataset on the dissolved organic carbon (DOC) and total nitrogen (TN) in ice cores from the TP yet. Even for the surface ice and snow, the data on microbial abundances, DOC, or TN have only been reported in less than ten glaciers (Hood et al., 2015; Liu et al., 2016b; Hu et al., 2018; Yan et al., 2020; Kang et al., 2022). Data scarcity significantly hindered our understanding of the biogeochemical cycle in TP glaciers and downstream ecosystems.

In this study, we aimed to construct a comprehensive dataset of microbial abundance, DOC, and TN in TP glaciers through extensive field sampling and experimental measurements. In this dataset, there are 5409 records from 12 glaciers for microbial abundance in ice cores and snow pits, and 2532 records from 38 glaciers for DOC and TN in the ice core, snow pit, surface ice, surface snow, and proglacial runoff. This dataset can provide fundamental data for analysing the storage, spatial pattern, and the drivers of glacier carbon and nitrogen on the TP. It can also facilitate the investigations of glacier biogeochemical cycles and evaluate the impact of glacier retreat on downstream ecosystems.

## 2 Study area

The TP covers an area of about 2.5 million $km^2$, with an average altitude of more than 4000 m above sea level. The climate over the TP is primarily influenced by the interaction between the Indian monsoon and the westerly wind (Tian et al., 2001). The TP and its surrounding regions contain the largest number of glaciers outside the poles with an area of about 47 thousand $km^2$ (Yao et al., 2012). These glaciers are important water resources for downstream areas and play a crucial role in regional water supply (Immerzeel et al. 2010). The glaciers on the TP are mainly distributed in the Kunlun, Nyainqntanglha, Himalayas, and Karakoram mountains, and most glaciers are located between 4500-6500 m above sea level (Liu et al., 2015). The glaciers can be divided into three types (Shi et al., 2000): monsoonal temperate glaciers (mainly distributed in the southeast of TP), subcontinental glaciers (mainly distributed in the northeast and southern margin of TP), and the continental glaciers (mainly distributed in the west of TP). Most of the TP glaciers except those in the Karakoram region are experiencing rapid retreat under climate warming (Yao et al., 2012; Wang et al., 2021).

## 3 Sample distribution and laboratory measurements

### 3.1 Sample distribution

In this study, 5409 microbial abundance data are distributed across 12 glaciers across the TP (Fig. 1a), including 5210 ice core samples from 7 glaciers and 199 snow-pit samples from 7 glaciers. For DOC and TN, 2532 samples from 38 glaciers were collected as shown in Fig. 1(b), including 1625 ice core samples from 7 glaciers, 180 surface ice samples from 17 glaciers, 100 snow pit samples from 4 glaciers, 254 surface snow samples from 28 glaciers, and 397 proglacial runoff samples from 16 glaciers.

The sampled glaciers cover diverse climate conditions. The multiyear average air temperature ranged from -13.4 °C (the Guliya glacier) to 2.9 °C (the Zhuxigou glacier), and the multiyear average precipitation ranged from 76.9 mm (the No.15 glacier) to 927.8 mm (the 24K glacier). The temperature and precipitation of most glaciers over the TP are within these ranges, which makes the dataset comprehensive and representative.

### 3.2 Sample distribution

Ice cores were drilled from the accumulation zones of nine glaciers to depths from 11 to 173 m (Fig. 1), and then transported frozen to the laboratory. Both microbial abundance and DOC/TN data were measured for samples from five glaciers (i.e. Muztagh Ata (MSTG), Cuopugou (CPG), Zuoqiupu (ZQP), Noijin Kangsang (NJKS), and East Rongbuk (ERB)); For samples from the Laohugou (LHG) and Geladandong (GLDD) glacier, only microbial abundance was measured; For samples from Muji (MJ) and Dunde (DD) glaciers, only DOC and TN were measured. The MJ, MSTG, LHG, DD, and GLDD glaciers are mainly influenced by the westerly, while the CPG, ZQP, NJKS, and ERB glaciers are strongly influenced by monsoon.

Snow pits were dug at the accumulation zones of seven glaciers, and snow was sampled from top to bottom at 5 or 10 cm intervals in each pit using a pre-sterilized steel scoop. Microbial abundance, DOC, and TN concentrations were measured for samples from GLDD, Zhadang (ZD), Palon No.4 (PL4), and ERB glacier, while only microbial abundance was measured for samples from DD, Mengdagangri (MDGR), and Yala glacier. More specifically, 14 snow pits were analysed for microbial abundance measurement, including six at ZD glacier during April, May, June, August, September, and October in 2006, two at the MDGR glacier in 2006 and 2007, two at different altitudes in the PL4 glacier, and one for each at the DD, GLDD, ERB, and Yala glacier, respectively. Four snow pits were sampled for DOC and TN concentration measurements with one snow pit in each glacier among GLDD, ZD, PL4, and ERB.

100

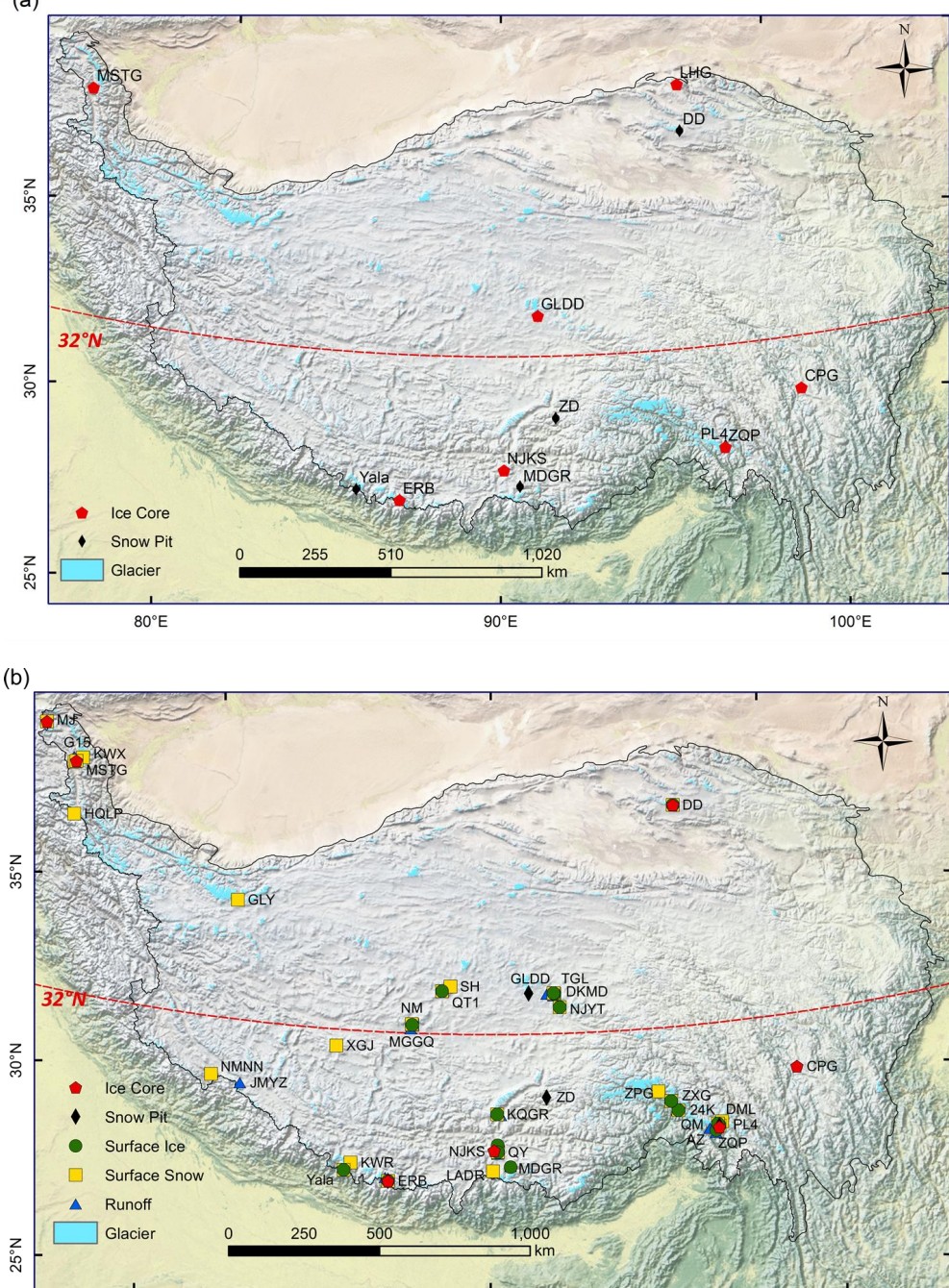

**Figure 1: Location of sampled glaciers for microbial abundance (a), DOC and TN (b) on the Tibetan Plateau. The abbreviations of glacier names were labelled in the map and the full names were available in the supplement.**

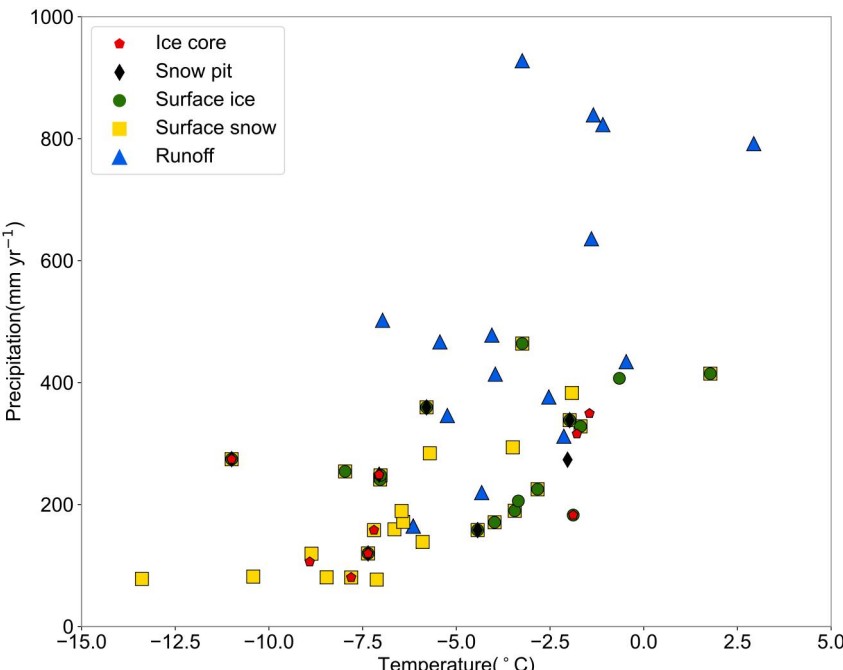

**Figure 2: Muti-year average air temperature and precipitation of sampled glaciers.**

Surface snow (top 10 cm) was sampled using a pre-sterilised steel scoop from the ablation and accumulation zones in 28 glaciers for DOC and TN concentration measurements. Surface ice samples were collected with a precleaned ice axe at the ablation zones in 17 glaciers. Ice and snow samples were stored in well-sealed WhirlPak bags (WhirlPak®, Nasco, USA) and transported to the laboratory under frozen conditions.

Proglacial runoff samples were collected from 16 glaciers using 100 mL polycarbonate bottles pre-cleaned using 1% HCl (Nalgene, USA). Totally eight sets of 24-h water samples were collected at six glaciers (i.e. MSTG, Tanggula Longxiazailongba (TGL), Qiangtang No.1 (QT1), Qiangyong (QY), Tonkmadi (DKMD), and PL4). The conductivity and pH of proglacial runoff were measured in the field with a YSI EXO2 Water Quality Sonde. Water samples were transported in a dark container with ice. In the laboratory, samples were frozen at -20°C until analysed.

**3.3 Laboratory measurements**

The ice cores were decontaminated by removing the outer 1-2 cm annulus with a knife, then were cut into 5-10 cm long sections and transferred into containers in a -20 °C clean room. Both knife and container have been pre-cleaned using 1% HCl and rinsed with filter water. Snow, ice, and ice core samples were melted overnight and the meltwater was aliquoted into 20 mL glass bottles. The glass bottles have been washed with 1% HCl, rinsed with deionized water three times, and then combusted (450 °C for > 3 h) before use.

Flow cytometry combined with the nucleic acid stain is a fast, accurate, quantitative, and reproducible technique for counting the number of bacteria (Hammes et al., 2008; Prest et al., 2013). The meltwater of snow, ice, and ice core samples were fixed with glutaraldehyde (final concentration: 1%) then stored at 4 °C, and were analysed within 8 hours after being stained with SYBR Green I (final concentration $1 \times 10^{-4}$, Marie et al., 1997). SYBR Green I is the standard dye used to distinguish microorganisms from abiotic particles in various environments (Van Nevel et al., 2017; Mao et al., 2022). Stained samples were filtered using a cell strainer (pore size 420 µm) before analysed using flow cytometry to prevent clotting of the machine. The signal of microorganisms was differentiated from the inorganic particles based on both fluorescence intensity and size (Prest et al., 2013). Samples were processed on an EPICS ALTRA II flow cytometer (Beckman Coulter, USA) (Liu et al., 2016a). Duplicate samples were measured with a relative standard deviation lower than 10%. Flow cytometry data were collected and analysed with CytoWin 4.31 software.

The DOC and TN concentrations were measured using a TOC-Lcph (Shimadzu Corp., Japan) following standard methods (Greenberg et al., 1992). Concentrations of $NO_3^-$ were measured using a Dionex ion Chromatograph System 2000 (Dionex Corp, USA) as previously described (Liu et al., 2016a).

## 4 Data description of microbial abundance

### 4.1 Snowpit

The microbial abundance in snow pits of the seven sampled glaciers (i.e.) ranged from 212 to 721 305 cells mL$^{-1}$, and the mean microbial abundances were 2 117, 8 664, 218 305, 12 479, 14 442, 64 515, and 12 401 cells mL$^{-1}$ for DD, GLDD, ZD, PL4, MDGR, Yala, and ERB, respectively. The range of these measurements is consistent with previous studies using the flow cytometer method (e.g. 3.7-25.0$\times10^4$ cells mL$^{-1}$ in the Kuytun 51 Glacier, Tianshan Mountains; Xiang et al., 2009; on the order of $10^3$ to $10^5$ cells mL$^{-1}$ in the alpine snowpack; Lazzaro et al., 2015; Fillinger et al., 2021). Fig. 3(a) shows the geospatial distribution of microbial abundance for the sampled snow pits. Generally, the microbial abundance is lower in the westerly dominated region than those in the monsoon dominated region. The DD glacier, located in the northeast of the TP, had the lowest microbial abundance (i.e. 2 177 cells mL$^{-1}$), while the ZD glacier, located in the south of the TP, had the highest abundance (i.e. 218 305 cells mL$^{-1}$), which was 100 times higher as that in DD. Figure 4 shows the variation of microbial abundance with depth in each snow pit, but no consistent patterns were observed.

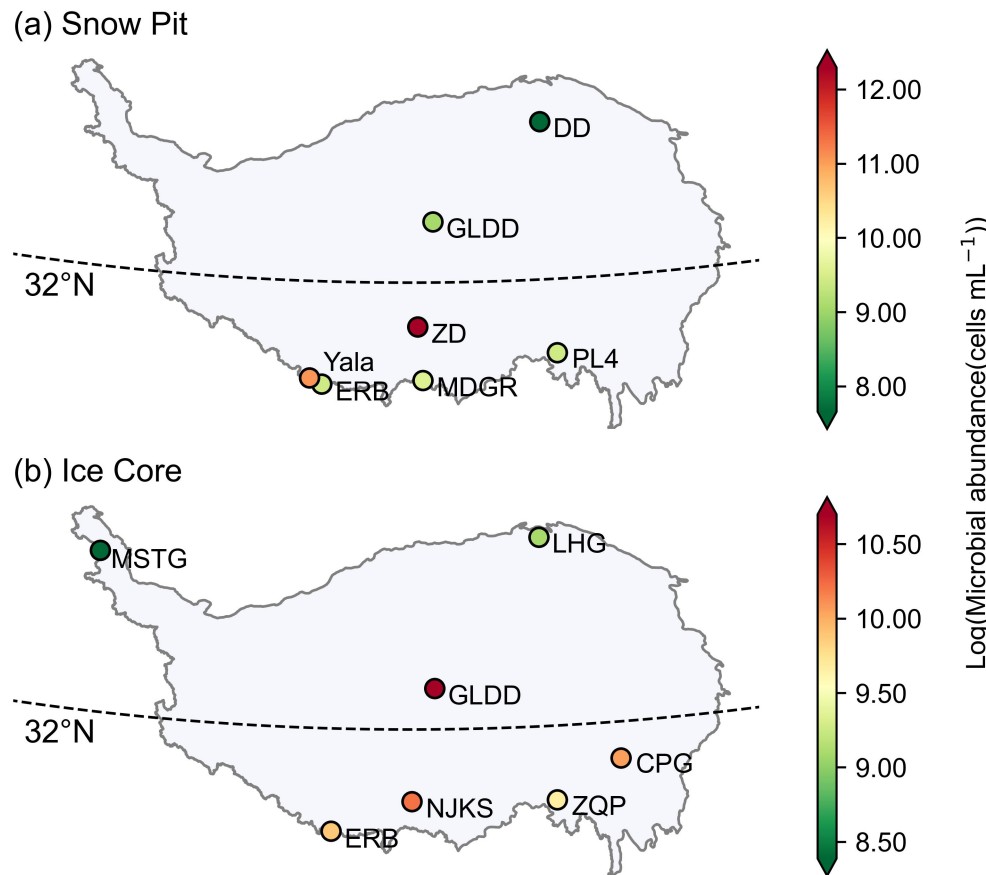

Figure 3: The spatial distribution of log(microbial abundance) for sampled snow pits (a) and ice cores (b) averaged in each glacier.

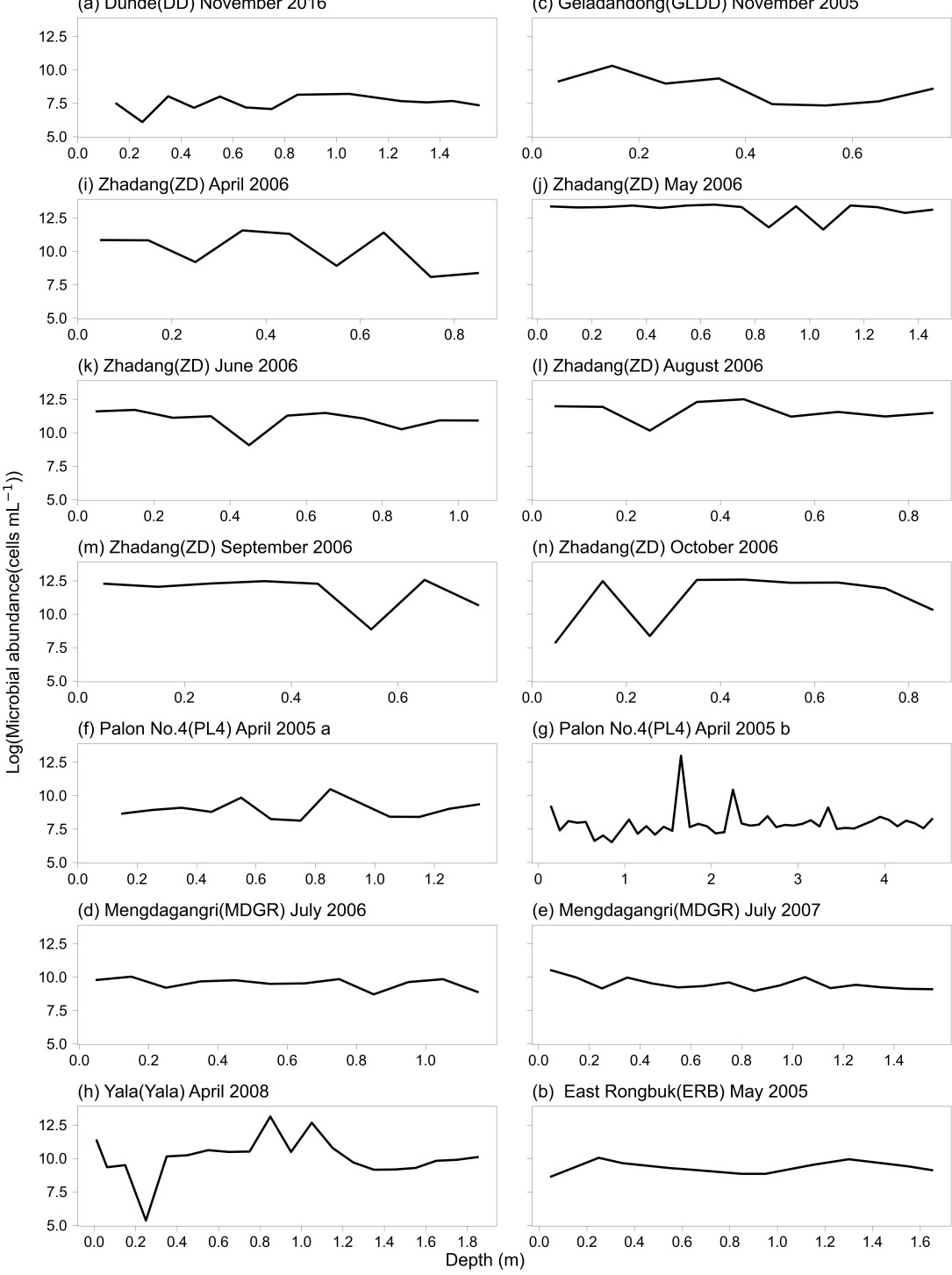

**Figure 4: The variation of microbial abundance with depth in each sampled snow pit. The abundance is log-transformed.**

## 4.2 Ice core

Microorganisms in glaciers are originated from atmospheric deposition, and it has been reported that microorganisms originating from the Saharan Desert have been found thousands of kilometers away in the Caribbean and European Alps (Kellogg et al., 2006). The deposited microorganisms are subjected to a range of post-depositional environmental selection processes (Chen et al., 2021), until they are buried by snow and eventually frozen in the ice core. The microbial abundance in ice cores of the seven sampled glaciers varied substantially, ranging from 63 to 1 130 080 cells mL$^{-1}$, and the mean microbial abundances were 4 389, 8 617, 44 318, 23 311, 15 648, 27 330, and 19 656 cells mL$^{-1}$ for MSTG, LHG, GLDD, CPG, ZQP, NJKS, and ERB, respectively. These values generally are in agreement with previous studies using the flow cytometer method (e.g. on the order of $10^4$ to $10^7$ in the GISP2 Greenland ice core; Miteva et al., 2009; $6.53 \times 10^3$ - $2.89 \times 10^5$ cells mL$^{-1}$ in the West Antarctic Ice Sheet Divide ice core; Santibanez et al., 2016). The spatial distribution pattern of microbial abundance in each ice core is shown in Fig. 3(b). Two ice cores in the north (i.e. MSTG and LHG) had a lower microbial abundance (i.e. 4 389 and 8 958 cells mL$^{-1}$, respectively) than those in the south where the microbial abundance was at least 15 688 cells mL$^{-1}$. Fig. 4 shows the variation of microbial abundance along with depth in each ice core. The microbial abundance of ice cores in CPG and NJKS glaciers had decreasing trends with depth, while that in the MSTG glacier had an increasing trend. There were low-frequency fluctuations observed for the microbial abundance in ice cores of ERB, GLDD, and LHG glaciers, and there were mainly high-frequency fluctuations in ZQP glacier.

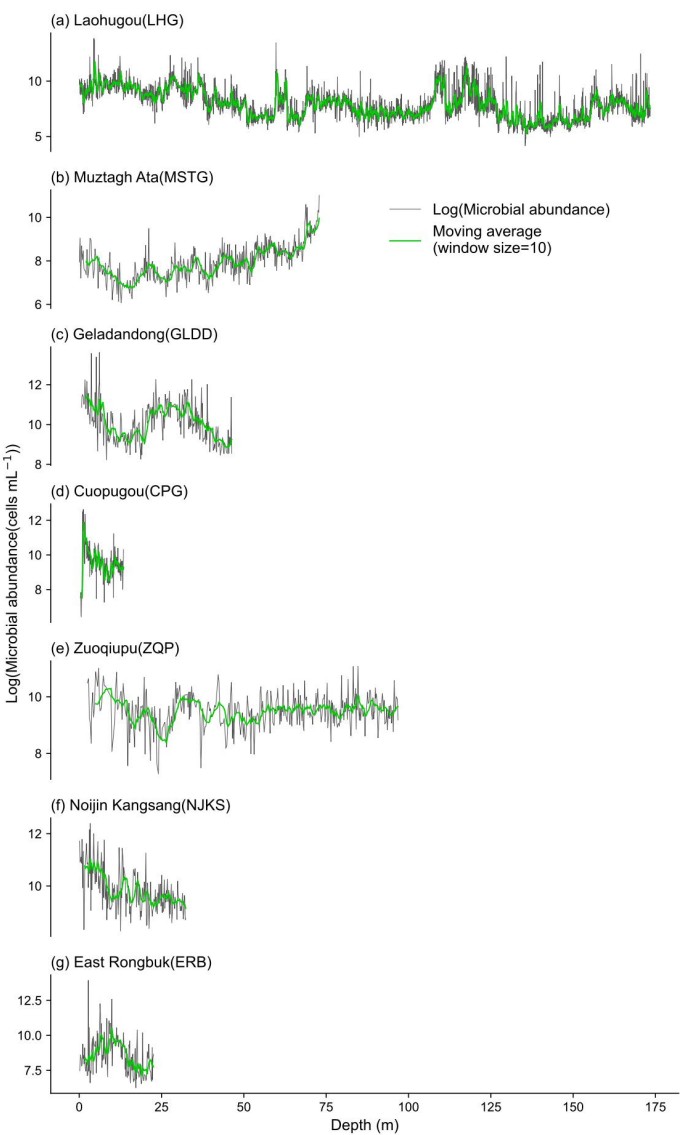

**Figure 5: The variation of microbial abundance along depth in seven ice cores. The window size used in the moving average method was 10. Microbial abundance is log-transformed.**

## 5 Data description of DOC and TN

 ### 5.1 Ice core

Organic carbon and nitrogen in ice cores can be from allochthonous or autochthonous sources. It has been reported that the wet DOC deposition ranges from 47 to 330 mg C m$^{-2}$ y$^{-1}$ (Yan et al., 2020) and the wet N deposition ranged from 44 to 155 mg N m$^{-2}$ y$^{-1}$ on the TP (Liu et al., 2015). In addition, microbial carbon fixation has also been reported in glacier surface

microbiome, and the average fixation rate in cryoconite holes of four glaciers on the TP is 1.77 μmol C m$^{-2}$ d$^{-1}$ (the yearly
rate is approximately 3.3 mg C m$^{-2}$ y$^{-1}$ assuming a growing season from May to September; Zhang et al., 2021), which is
substantially lower than the atmospheric deposition rate. The microbial nitrogen fixation rate has not been quantified, but a
study in the Arctic region has been reported as 0.04 mg N m$^{-2}$ y$^{-1}$ (Telling et al., 2011), which is again orders of magnitude
lower than the atmospheric deposition.

The DOC concentrations in ice core samples ranged from 0.005 to 5.05 mg L$^{-1}$ with an average value of 0.54±0.38 mg L$^{-1}$.
These values are larger than the englacial DOC concentrations in global mountain glaciers reported by Hood et al. (2015)
(0.01 to 1.20 mg L$^{-1}$, 0.29±0.03 mg L$^{-1}$ on average), which may be related to the higher aerosol concentration of TP
(Spracklen et al., 2011). The TN concentrations ranged from 0.001 to 1.15 mg L$^{-1}$ with an average value of 0.24±0.16 mg L$^{-1}$.
Fig. 6 shows the variation of DOC and TN concentrations across the vertical profiles in ice cores of the MSTG and ERB
glaciers. Generally, there are decreasing trends of DOC and TN with depth, suggesting that atmospheric deposition has been
increasing in recent years. There exist large inter-annual variations in the serial data with some large values occasionally,
which may be related to historical large sand storms.

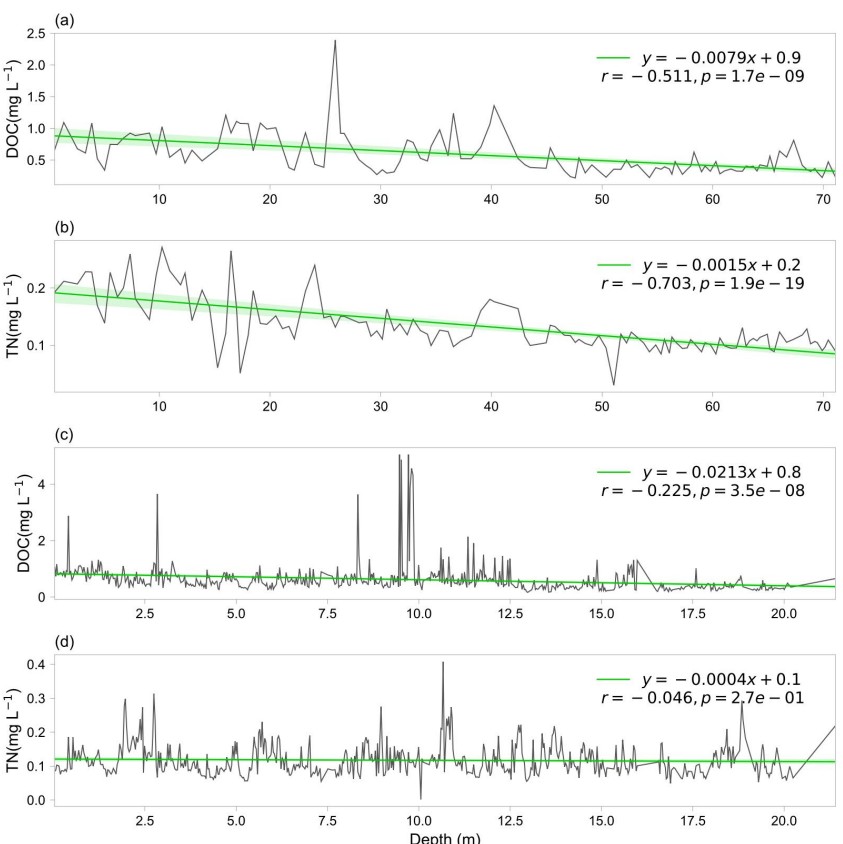

**Figure 6: Variation of DOC (a, c) and TN (b, d) concentrations along vertical profiles in ice cores of the MSTG (Muztagh Ata) glacier (a, b) and the ERB (East Rongbuk) glacier (c,d).**

## 5.2 Surface ice, surface snow, and snow pit

The DOC concentrations ranged from 0.08 to 9.0 mg L$^{-1}$ (0.89±1.05 mg L$^{-1}$), from 0.12 to 11.65 mg L$^{-1}$ (1.19±1.78 mg L$^{-1}$), and from 0.05 to 16.15 mg L$^{-1}$ (0.72±1.71 mg L$^{-1}$) in surface ice, surface snow, and snow pits, respectively. These values are comparable to the DOC concentration of surface ice in four TP glaciers reported by Liu et al. (2016b) (i.e. 1.01±0.22 mg L$^{-1}$), and those in surface snow (mean values ranging from 0.16 to 1.17 mg L$^{-1}$) and snow pits (mean values ranging from 0.21 to 0.81 mg L$^{-1}$) in TP glaciers (Gao et al., 2020). The TN concentration ranged from 0.01 to 1.88 mg L$^{-1}$ (0.19±0.22 mg L$^{-1}$), from 0.07 to 3.06 mg L$^{-1}$ (0.34±0.35 mg L$^{-1}$), and from 0.02 to 0.84 mg L$^{-1}$ (0.15±0.15 mg L$^{-1}$) in surface ice, surface snow, and snow pits, respectively. Fig. 7 shows the spatial distribution of DOC and TN concentrations for surface snow and surface ice. For the DOC concentrations of surface snow, there is a decreasing trend from south to north in the monsoon dominated region (i.e. located to the south the 32°N). The DOC and TN concentrations in the westerly- and monsoon-dominated regions were not significantly different (Mann Whitney U test, *P = 0.95*). The differences in TN concentration between the two regions were evident with higher values identified in the westerly dominated region although the Mann Whitney U test was not significant with a p-value of 0.056.

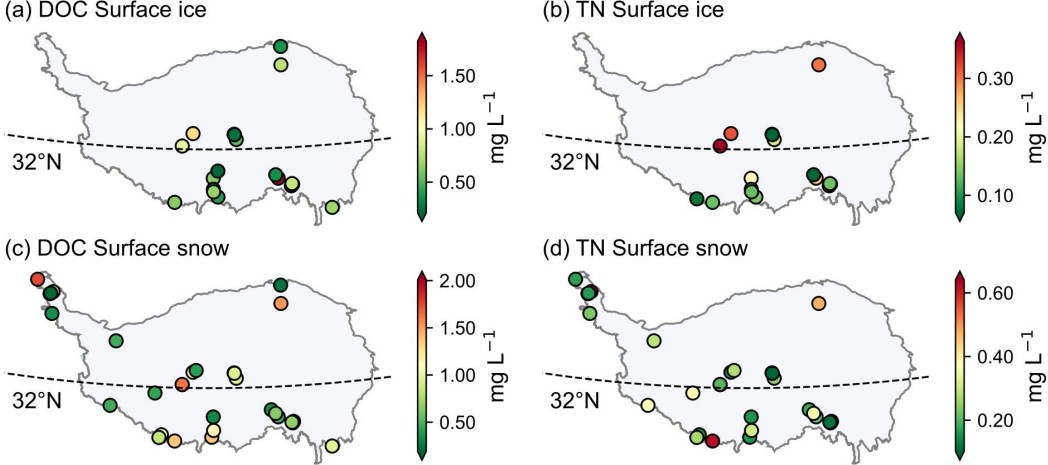

**Figure 7: The spatial distribution of DOC (a, c) and TN (b, d) concentrations for surface ice (a, b) and surface snow (c, d).**

## 5.3 Runoff

The DOC concentration ranged from 0.16 to 4.94 mg L$^{-1}$ (0.79±0.68 mg L$^{-1}$) in the proglacial runoff, which is consistent with those observed worldwide (42 glaciers, 0.10 - 3.40 mg L$^{-1}$; Li et al., 2017). The TN concentration ranged from 0.05 to 2.3 mg L$^{-1}$ (0.29±0.22 mg L$^{-1}$). The runoff is neutral to alkaline with the pH value ranging from 7.3 to 12.4 (9.10±0.88). Fig. 8 shows the spatial distribution of DOC, TN, NO$_3^-$ concentrations, and pH for the proglacial runoff on the TP. For the DOC concentration of runoff, there was a decreasing trend from west to east. There were significant differences in the concentrations of TN and NO$_3^-$ in the runoff between the westerly and monsoon regions with p-values both less than 0.01 (Mann Whitney U test) and the values in the westerly dominated region were higher. The pH value was also larger in the

westerly dominated region than in the monsoon dominated region although the Mann Whitney U test was not significant with a p-value of 0.07.

Fig. 9 shows the diurnal variation of DOC, TN, NO₃⁻, and conductivity in proglacial runoff of the Qiangyong glacier. No obvious pattern in the diurnal curve of DOC was observed, while there were obvious unimodal patterns for TN, NO₃⁻, and conductivity. These observations are very helpful for biogeochemical studies at fine temporal resolutions.

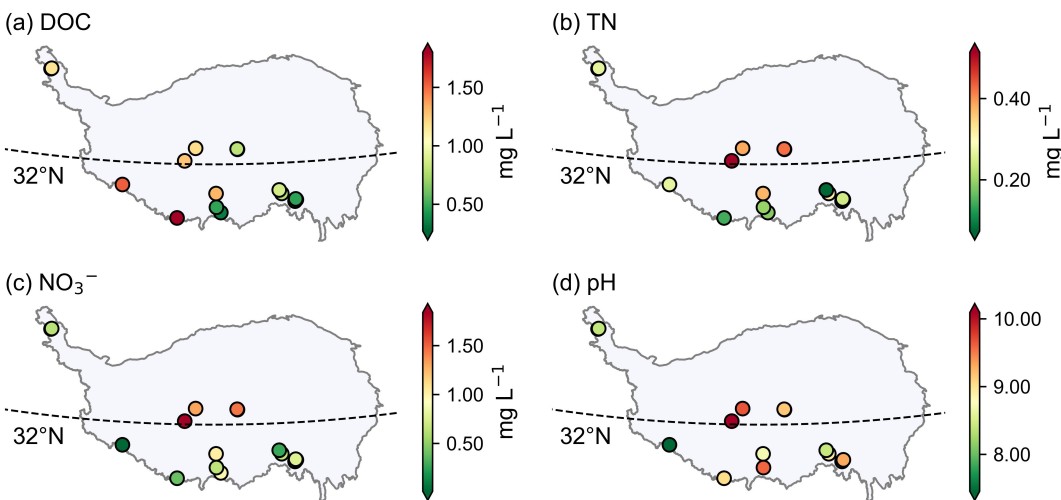

**Figure 8: The spatial distribution of DOC (a), TN (b), NO₃⁻ (c) concentrations and pH (d) for proglacial runoff on the**
220 **Tibetan Plateau.**

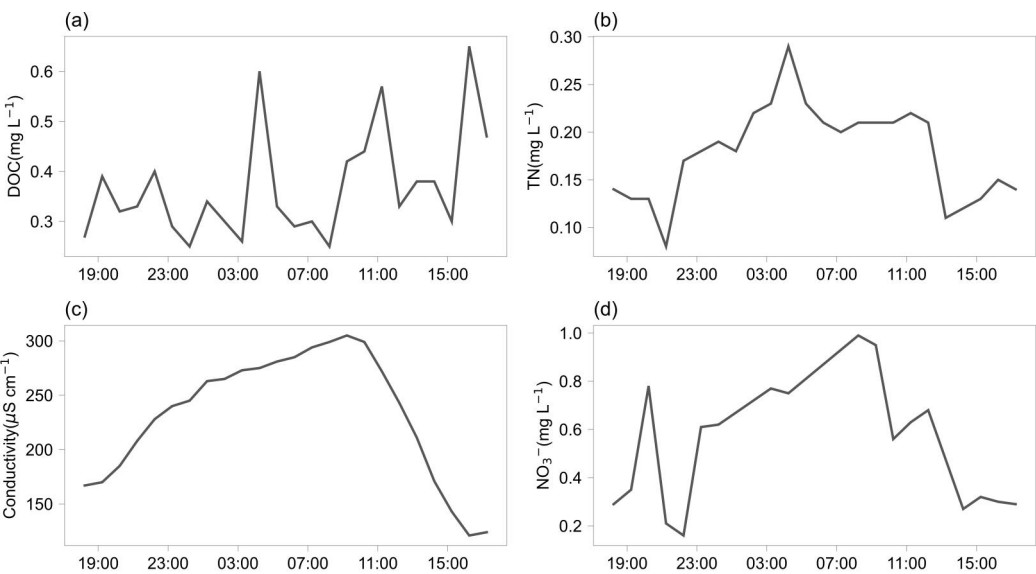

**Figure 9: The 24-h time series of DOC (a), TN (b), conductivity (c), and NO₃⁻(d) in proglacial runoff of the Qiangyong glacier.**

## 5.4 Comparison among different habitats

The DOC and TN concentrations were compared among the various glacial habitats (ice core, snow pits, surface ice, surface snow, and proglacial runoff). The result showed that the DOC concertation in ice core was significantly lower than that in surface ice (Mann-Whitney-Wilcoxon test, $P$ = 0.02), while the differences of TN in ice core and surface ice were not significant ($P$ = 0.24). The DOC and TN concentrations were also lower in snow pits than in surface snow. The difference was not significant for DOC with a p-value of 0.19 and was significant for TN with a p-value of 0.03. The DOC and TN concentrations in proglacial runoff were similar to those in surface snow (Mann Whitney U test, $P$ = 0.95 and 1.0 for DOC and TN, respectively).

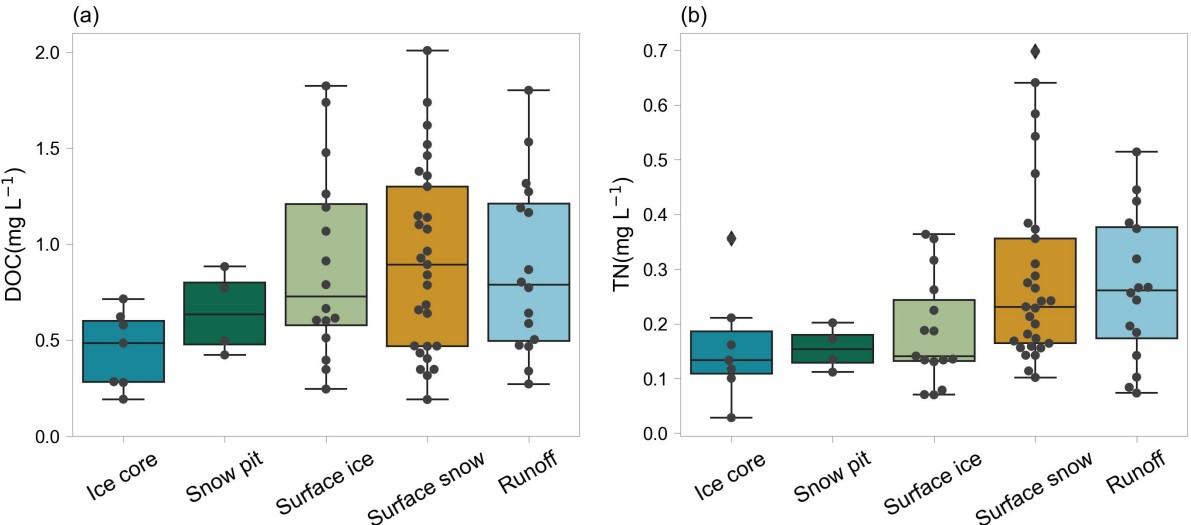

**Figure 10: The boxplots of the DOC (a) and TN (b) concentrations in five main glacial habitats on the Tibetan Plateau.**

## 6 Data availability

The dataset of microbial abundance, DOC, and TN in the ice core, snow pit, surface ice, and snow from the Tibetan Plateau glaciers are accessible at the National Tibetan Plateau/Third Pole Environment Data Center (https://doi.org/10.11888/Cryos.tpdc.271841, Liu, 2021).

## 7 Conclusions

We constructed a dataset of microbial abundance, DOC, and TN for glaciers on the TP. The dataset comprises 5 409 microbial abundance data from 12 glaciers and 2 532 DOC and TN data from 38 glaciers. The sampled glaciers cover diverse geographic and climatic regions, which makes the dataset qualified for large-scale studies across the TP. This

systematic dataset can provide important information on carbon and nitrogen cycles in glacial ecosystems. It can be used to evaluate the response of carbon and nitrogen cycling to global climate change, and to estimate the impact of glacier melting on downstream ecosystems such as glacier-feed streams and lakes. The time-series data of microbial abundance in ice cores can be used as an indicator of past climate change, and the spatial distribution of DOC and TN data can be used to estimate the storage and spatial distribution of glacier carbon and nitrogen, which are essential inputs for biogeochemical models of glacial ecosystems. Considering its broad spatial and temporal coverage, this dataset can serve as an important data source for forecasting the impact of warming on glacial carbon and nitrogen cycles at regional and even global scales.

**Author contributions**

YL and JL designed the study and wrote the manuscript. PF and JL compiled and analysed the dataset. YL, BG, MJ, PL, GM, BX, and SK performed field sampling and experimental measurement. All authors contributed to the writing and editing of this paper.

**Competing interests**

The authors declare that they have no conflict of interest.

**Financial support**

This work was supported by the National Key Research and Development Program of China (2019YFC1509103), the Second Tibetan Plateau Scientific Expedition and Research program (2019QZKK0503), and the National Natural Science Foundation of China (U21A20176 and 42171132).

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
