# Peer review of "A comprehensive dataset of microbial abundance, dissolved organic carbon, and nitrogen in Tibetan Plateau glaciers"

_Earth System Science Data, 2022_

## Author Comment (AC1)

**RC1: 'Comment on essd-2022-10', Arwyn Edwards**

This paper describes a large and unique collection of microbial abundance, DOC, nitrogen and phosphorus data across a globally-relevant region of glacial ecosystems. It will provide a valuable resource. Overall, my recommendation is to provide enhanced discussion of ecological insights, comparison with other studies, including in other regions and to provide a cautious treatment of the analytical methodologies used.

Reply: Dear Arwyn, thank you very much for the positive comments and helpful suggestions. According to your suggestions, the discussions of ecological insights and the comparison with other studies have been enhanced, and the descriptions of analytical methodologies (e.g. that about the flow cytometry) have also been improved. The details are described as follows.

Key areas for improvement

(1) While this is a data description paper, some additional context in the discussion on the ecological insights which might be permitted from these datasets would be valuable

Reply: The discussions on the ecological insights of our dataset have been enhanced in the introduction and conclusion part.

L53-L57: "we aim to construct a comprehensive dataset of microorganisms, dissolved organic carbon (DOC), and total nitrogen (TN) in TP glaciers by compiling deep ice-core samples and extensive surface snow and ice samples, which on the one hand can provide fundamental data for analysing the storage, spatial pattern and related drivers of glacier carbon and nitrogen on the TP, and on the other hand can facilitate the researches on glacier biogeochemical cycle and the impact of glacier retreat on downstream ecosystems."

L236-L243: "This systematic dataset can provide important information for the studies on carbon and nitrogen cycle in glacial ecosystems, their response to global climate change, and their impact on downstream ecosystems such as glacier-feed streams and lakes. The time series data of microbial abundance in ice cores can be used as an indicator of past climate change, and the spatial distribution of DOC and TN data can be used to estimate the storage and spatial distribution of glacier carbon and nitrogen, which are essential inputs for biogeochemical models of the glacial ecosystems. Considering its broad spatial and temporal coverage, this dataset can

serve as an important data source for forecasting the impact of warming on glacial carbon cycle at regional and even global scales."

(2) How does the dataset compare with other studies of microbial abundance and biogeochemical parameters in the Tibetan Plateau glaciers and beyond?

Reply: The observations of microbial abundance and DOC in this study are generally consistent with existing studies in the Tibetan Plateau (TP) glaciers and beyond. For example, the range of microbial abundance in snow pits (i.e. 212 to 721 305 cells mL$^{-1}$) was consistent with the results of existing researches using the flow cytometer method (e.g. 3.7-25.0×10$^4$ cells mL$^{-1}$ in the Kuytun 51 Glacier, Tianshan Mountains; Xiang et al., 2009; on the order of 10$^3$ to 10$^5$ cells mL$^{-1}$ in the alpine snowpack; Lazzaro et al., 2015; Fillinger et al., 2021). The mean DOC values in this study were 0.89±1.05 mg L$^{-1}$, 1.19±1.78 mg L$^{-1}$, and 0.72±1.71 mg L$^{-1}$ in surface ice, surface snow, and snow pits, respectively. These values are comparable to the DOC concentrations of surface ice in four TP glaciers reported by Liu et al. (2016) (i.e. 1.01±0.22 mg L$^{-1}$), and those of surface snow (mean values ranging from 0.16 to 1.17 mg L$^{-1}$) and snow pits (mean values ranging from 0.21 to 0.81 mg L$^{-1}$) in TP glaciers summarized by Gao et al. (2020). The discussions about these comparisons have been enhanced in the revised manuscript (L135-137, L154-156, L177-179, and L189-192).

In addition, to the best of our knowledge, this is the first publicly available dataset of microbial abundance and TN in TP glaciers and also the first dataset of DOC in ice cores on the TP.

References:

Fillinger, L., Hürkamp, K., Stumpp, C., Weber, N., Forster, D., Hausmann, B., Schultz, L., and Griebler, C.: Spatial and Annual Variation in Microbial Abundance, Community Composition, and Diversity Associated With Alpine Surface Snow, Front. Microbiol., https://www.frontiersin.org/article/10.3389/fmicb.2021.781904, 2021.

Gao, T., Kang, S., Zhang, Y., Sprenger, M., Wang, F., Du, W., Wang, X., and Wang, X.: Characterization, sources and transport of dissolved organic carbon and nitrogen from a glacier in the Central Asia, Sci. Total Environ., 725, 138346, https://doi.org/https://doi.org/10.1016/j.scitotenv.2020.138346, 2020.

Lazzaro, A., Wismer, A., Schneebeli, M., Erny, I., and Zeyer, J.: Microbial abundance and community structure in a melting alpine snowpack, Extremophiles, 19, 631–642, https://doi.org/10.1007/s00792-015-0744-3, 2015.

Liu, Y., Xu, J., Kang, S., Li, X., and Li, Y.: Storage of dissolved organic carbon in Chinese glaciers, J. Glaciol., 62, 402-406, https://doi.org/10.1017/jog.2016.47, 2016.

Xiang, S.-R., Shang, T.-C., Chen, Y., Jing, Z.-F., and Yao, T.: Changes in diversity and biomass of bacteria along a shallow snow pit from Kuytun 51 Glacier, Tianshan Mountains, China, J. Geophys. Res. Biogeosciences, 114, https://doi.org/10.1029/2008JG000864, 2009.

(3) Flow cytometry is a key method in the paper and has been used for over a decade to enumerate microbes in glacial samples. It has advantages, but also some limitations, for example in the discrimination between biological cells and inorganic particulates, which may also fluoresce. The manuscript would benefit from a discussion of the pros and cons of flow cytometry in this context, and whether potential sources of interference were mitigated in this study.

Reply: The discussion of the pros and cons of flow cytometry and how the potential sources of interference were mitigated in this study have been added (L117-124).

"Flow cytometry combined with the nucleic acid stain is a fast, accurate, quantitative and reproducible technique for counting the number of bacteria (Hammes et al., 2008; Prest et al., 2013), which was used for the enumeration of bacteria in this study. 1.98 mL of meltwater was fixed with glutaraldehyde (final concentration: 1%), stored at 4 °C, and analysed within 8 hours after staining with SYBR Green I (Marie et al., 1997). SYBR Green I is the standard dye used in the analysis of various environments to distinguish bacteria from abiotic particles (Van Nevel et al., 2017; Mao et al., 2022). Staining could capture inorganic particulates and result in false positives, which were mitigated by the following experimental controls in this study:1) if particulates were large, they would have been removed in the filtering step before analysis; 2) fixed gating could distinguish inorganic background and bacteria (Prest et al., 2013)."

**References:**

Hammes, F., Berney, M., Wang, Y., Vital, M., Köster, O., and Egli, T.: Flow-cytometric total bacterial cell counts as a descriptive microbiological parameter for drinking water treatment

processes, Water Res., 42, 269–277, https://doi.org/https://doi.org/10.1016/j.watres.2007.07.009, 2008.

Mao, G., Ji, M., Xu, B., Liu, Y., and Jiao, N.: Variation of High and Low Nucleic Acid-Content Bacteria in Tibetan Ice Cores and Their Relationship to Black Carbon, Front. Microbiol., 13, https://www.frontiersin.org/article/10.3389/fmicb.2022.844432, 2022.

Prest, E. I., Hammes, F., Kötzsch, S., van Loosdrecht, M. C. M., and Vrouwenvelder, J. S.: Monitoring microbiological changes in drinking water systems using a fast and reproducible flow cytometric method, Water Res., 47, 7131-7142, https://doi.org/https://doi.org/10.1016/j.watres.2013.07.051, 2013.

Van Nevel, S., Koetzsch, S., Proctor, C. R., Besmer, M. D., Prest, E. I., Vrouwenvelder, J. S., Knezev, A., Boon, N., and Hammes, F.: Flow cytometric bacterial cell counts challenge conventional heterotrophic plate counts for routine microbiological drinking water monitoring, Water Res., 113, 191-206, https://doi.org/https://doi.org/10.1016/j.watres.2017.01.065, 2017.

L31: microorganisms

Reply: We 've corrected the typo (L31).

L35:  Egge et al., 2021 - this marine-based paper seems at odds with the claims around glacial microbes and carbon cycles, and is formatted incorrectly in the references. I suggest that more relevant citations are provided to support this claim.

Reply: We have replaced the citation "Egge et al., 2021" with "Smith et al., 2017; Irvine-Fynn et al., 2021" which are more relevant to glacial microbes and carbon cycles.

References:

Irvine-Fynn, T. D. L., Edwards, A., Stevens, I. T., Mitchell, A. C., Bunting, P., Box, J. E., Cameron, K. A., Cook, J. M., Naegeli, K., Rassner, S. M. E., Ryan, J. C., Stibal, M., Williamson, C. J., and Hubbard, A.: Storage and export of microbial biomass across the western Greenland Ice Sheet, Nat. Commun., 12, 3960, https://doi.org/10.1038/s41467-021-24040-9, 2021.

Smith, H. J., Foster, R. A., McKnight, D. M., Lisle, J. T., Littmann, S., Kuypers, M. M. M., and Foreman, C. M.: Microbial formation of labile organic carbon in Antarctic glacial environments,

10, 356–359, https://doi.org/10.1038/ngeo2925, 2017.

L126: Considering the range of sites, conditions and sample types, is one overall mean value appropriate?

Reply: According to the suggestion, the one overall mean value has been removed, and the mean value in each site has been added (L134-135 and also L152-153).

L127: The range of these measurements was consistent with the results of existing researches using the flow cytometer method - how do they compare with flow cytometric analyses from other regions, e.g. Svalbard, Greenland, Antarctica?

Reply: Thanks for the suggestion. After careful literature review, we found there is limited literature on the use of flow cytometry to measure bacterial count in snows in Svalbard, Greenland and Antarctica regions. There were some researches on alpine snowpack using the flow cytometry method (Lazzaro et al., 2015; Fillinger et al., 2021), and the cell counts of snow pits in this study were on the same order as these researches. This has been explained in L135-137.

References:

Lazzaro, A., Wismer, A., Schneebeli, M., Erny, I., and Zeyer, J.: Microbial abundance and community structure in a melting alpine snowpack, Extremophiles, 19, 631–642, https://doi.org/10.1007/s00792-015-0744-3, 2015.

Fillinger, L., Hürkamp, K., Stumpp, C., Weber, N., Forster, D., Hausmann, B., Schultz, L., and Griebler, C.: Spatial and Annual Variation in Microbial Abundance, Community Composition, and Diversity Associated With Alpine Surface Snow, Front. Microbiol., https://www.frontiersin.org/article/10.3389/fmicb.2021.781904, 2021.

---

## Author Comment (AC2)

**RC2: 'Comment on essd-2022-10', Anonymous Referee #2**

General comments

This manuscript shows the dataset of microbial abundance and geochemistry of ice cores, snow pit, surface ice/snow, and glacial runoff collected from 40 mountain glaciers in Tibetan Plateau. The data set contains valuable information on microbes and chemistry to study glacier ecosystems of the region. Since the distinct microbial communities from polar regions have been reported on Asian glaciers, it is important to publish such data set. Although some additional information is necessary in the data set as shown below, I would support to publish them after revision.

Reply: We appreciate the reviewer's time and positive comments on our manuscript.

L75 Does "the multiyear average temperature" mean air temperature or snow temperature? Please specify. If snow/ice temperature is available for all glaciers, please add them in "Glacier info.xlsx".

Reply: Thanks for the nice suggestion! We have rephrased "the multiyear average temperature" to "the multiyear average air temperature" to make it clear (L21, L76, and L102), and we do not have snow/ice temperature for all glaciers.

L80 There is a lack of elevation of the sites of sample collections as microbes and snow/ice chemistry vary with elevation. Please indicate the locations of the sites of the ice cores and snow pits.

Reply: The longitude, latitude, elevation, and sampling time of ice cores and snow pits have been added in the "Icecore-info.xlsx" and "Snowpit-info.xlsx" files. The "IcecoreID" or "SnowpitID" field can be used to link these information to the records in "Microbial abundance-ice core.xlsx", "Microbial abundance-snow pit.xlsx", "DOC-TN-ice core.xlsx", and "DOC-TN-sonw pit.xlsx". These files have been updated in https://doi.org/10.11888/Cryos.tpdc.271841.

L121-123 There is a lack of measurement procedures for conductivity and pH shown in Figure 9.

Reply: The conductivity and pH of proglacial runoff were measured with the YSI EXO2 Water Quality Sonde. This has been added in Section 3.3 (L129-130).

L140-141 I wonder the bacteria analyzed in this study were those grew in situ in snow or were cells deposited from atmosphere. Please explain the possible sources of bacteria in each ice core.

Reply: The possible sources of bacteria in ice cores have been added in L148-151.

"Bacteria in glacier are originated from atmospheric deposition, and it has been reported that microorganisms originating from the Saharan Desert have been found thousands of kilometers away in the Caribbean and European Alps (Kellogg et al., 2006). The deposited microorganisms are subjected to a range of post-depositional environmental selection processes (Chen et al., 2021), until they are buried by snow and eventually frozen in the ice core."

L146 Is there any geographical trend of the bacterial abundance? Also, is there any relationship between the altitude of drilling site and the bacterial abundance?

Reply: At the regional scale, the bacterial abundance of both ice cores and snow pits in the north of Tibetan Plateau was generally higher than that in the south of Tibetan Plateau. This has been described in 139-141 to L157-159.

As there were no altitude gradients in the ice-core or snow-pit samples in each glacier, the relationship between the altitude of drilling site and the bacterial abundance was not analyzed.

L155 It would be worth to add some explanation of possible sources of DOC and TN in the ice cores.

Reply: The possible sources of DOC and TN in the ice cores have been added in L168-175.

"Organic carbon and nitrogen in ice cores can be both from both allochthonous or autochthonous sources. It has been reported that the wet DOC deposition ranged from 47 to 330 mg C m$^{-2}$ y$^{-1}$ (Yan et al., 2020) and the wet N deposition ranged from 44 to 155 mg N m$^{-2}$ y$^{-1}$ on the TP (Liu et al., 2015). In addition, microbial carbon fixation has also been reported in glacier surface microbiome, and the average fixation rate in cryoconite holes of four glaciers on the TP was 1.77 μmol C m$^{-2}$ d$^{-1}$ (the yearly rate was approximately 3.3 mg C m$^{-2}$ y$^{-1}$ assuming a growing season from May to September) (Zhang et al., 2021), which is substantially lower than the atmospheric deposition rate. The microbial nitrogen fixation rate has not been quantified, but a research at the Arctic region has been reported as 0.04 mg N m$^{-2}$ y$^{-1}$ (Telling et al., 2011), which is again orders of

magnitude lower than the atmospheric deposition. ”

References:

Liu, Y. W., Xu-Ri, Wang, Y. S., Pan, Y. P., and Piao, S. L.: Wet deposition of atmospheric inorganic nitrogen at five remote sites in the Tibetan Plateau, Atmos. Chem. Phys., 15, 11683-11700, https://doi.org/10.5194/acp-15-11683-2015, 2015.

Telling, J., Anesio, A. M., Tranter, M., Irvine-Fynn, T., Hodson, A., Butler, C., and Wadham, J.: Nitrogen fixation on Arctic glaciers, Svalbard, J. Geophys. Res. Biogeosciences, 116, https://doi.org/https://doi.org/10.1029/2010JG001632, 2011.

Yan, F., Wang, P., Kang, S., Chen, P., Hu, Z., Han, X., Sillanpää, M., and Li, C.: High particulate carbon deposition in Lhasa—a typical city in the Himalayan–Tibetan Plateau due to local contributions, Chemosphere, 247, 125843, https://doi.org/https://doi.org/10.1016/j.chemosphere.2020.125843, 2020.

Zhang, Y., Kang, S., Wei, D., Luo, X., Wang, Z., and Gao, T.: Sink or source? Methane and carbon dioxide emissions from cryoconite holes, subglacial sediments, and proglacial river runoff during intensive glacier melting on the Tibetan Plateau, Fundam. Res., 1, 232–239, https://doi.org/https://doi.org/10.1016/j.fmre.2021.04.005, 2021.

More information on the glaciers would be worth in "Glacier info.xlsx". For example, mountain range of the location, type of glaciers (valley or ice cap), elevation range, equilibrium line altitude (ELA).

Reply: More information has been added to "Glacier info.xlsx" as suggested, including the mountain range (the "MountainName" field), the glacier types by geomorphology (the "GlacierTypeGeomorph" field) and by climate (the "GlacierTypeClim" field), elevation range (the "ElevMin(m)" and "ElevMax(m)" fields), and equilibrium line altitude (the "ELA" field). The file has been updated in https://doi.org/10.11888/Cryos.tpdc.271841.

Date of sampling, coordinates, and elevation are necessary for each sample in "DOC-TN surface snow ice.xlsx" as geochemistry of the surface snow and ice varies temporally and spatially.

Reply: The information of sampling date (the "SamplingTime" field), coordinates (the "Longitude" and "Latitude" field) , and elevation (the "Elevation(m)" field) has been added to the "DOC-TN

surface snow ice.xlsx" when available. In addition, the elevation extracted from DEM (Digital

Elevation Model) was also added (the "Elevation_fromDEM(m)" field) considering the in-situ

elevation of some sites was not measured. The file has been updated in

https://doi.org/10.11888/Cryos.tpdc.271841.

As water characteristics of glacier runoff have a diurnal variation, it is necessary to show the time

of sample correction for the runoff data ("DOC-TN-runoff.xlsx"). The time seems to be partially

included in the column of "SampleID", but they would be better to be shown in an independent

column. Time zone (probably Beijing standard time?) should also be shown.

Reply: The information of sampling time and time zone (i.e. the "SamplingTime" and "TimeZone"

fields) has been added to the runoff data. The file has been updated in

https://doi.org/10.11888/Cryos.tpdc.271841.

---

## Author Response (AR2)

Comments to the author:

The present version, with bits of coloured text, does not represent a final product? Please ensure that publisher begins next steps starting from best version of manuscript.

Reply: Thanks for your kind reminder. We have double checked that the submitted version is the best version of our manuscript.

Translation glitches remain. Very good language services from Copernicus should help solve most of those but authors will need to examine proofs very carefully. Add an English-speaker to your proof team?

Reply: The revised manuscript has been polished throughout by an English-speaker.

Line 65: "marine" glaciers in TP? Technically, marine glaciers terminate at sea level in salt water. Do you mean instead glaciers influenced by marine-derived moisture flows?

Reply: Yes, we mean "glaciers influenced by marine-derived moisture flows" by using "marine glaciers". In the revised manuscript, "marine glaciers" has been rephrased to "monsoonal temperate glaciers" (L67).

Thank you for patience. Thank you again for using ESSD.

Reply: Thank you very much for your time and help!